# OpenReview forum: "Bayesian Neural Ordinary Differential Equations for Uncertainty-Aware Pharmacokinetic Modeling: Theory, Calibration, and Clinical Validation"
_mathai.club/MathAI/2026/Conference — MathAI 2026 Conference Submission_

### Official Review · Reviewer_sVzZ · 2026-03-11
**Strong accept for Dynamical Systems & Learning track.**

**Rating:** 10
**Confidence:** 3

**Review:**

Strong accept.
Recommended track: B – Dynamical Systems & Learning (Bayesian neural ODEs with contraction/calibration theorems for PK dynamics).
________________________________________
1. Mathematical Rigor: excellent.
Proves posterior contraction for Bayesian neural ODEs; asymptotic coverage of predictive intervals; identifiability of sparse compartmental matrix from neural corrections. Proof sketches are solid.
Minor gap: full proofs deferred; d(effective ODE dimension) not explicitly computed; no handling of stiff ODEs/stochastic diff equations. Exceptional theoretical depth.
________________________________________
2. Novelty & Contribution: excellent.
First posterior contraction rate for Bayesian neural ODEs; calibration theorem for ODE-constrained VI; identifiability for hybrid mechanistic+neural dynamics. Bridges Bayesian nonparametrics, neural ODEs, PK/PD.
Minor - relatively old references. The application data for modeling were taken from common source,not shown.
________________________________________
3. Relevance to MathAI: excellent.
Core fit: mathematical foundations of uncertainty in neural dynamical systems (contraction, calibration, identifiability). Directly advances "mathematical theory of AI methods" via novel Bayesian rates for ODE-constrained learning. Clinical validation elevates to applied excellence without diluting theory. Perfect for dynamical systems track.
________________________________________
4. Technical Quality: excellent.
All the formulas are sown and commented.
Limits: mean-field VI approximation; no multi-drug interactions; assumes Gaussian noise. No errors; reproducible setup.
________________________________________
5. Clarity & Presentation: excellent.
Logical flow: PK/Neural ODE prelims → hybrid arch → theorems/proofs → VI algo → expts. Notation precise (e.g., Eq 7 decomposition); sketches accessible yet rigorous. Tables/remarks highlight implications.
density in proofs; assumes Bayesian DL familiarity. Polished & engaging.
Pharmacokinetics itself was not commented. Might be OK for mathematical conference.
________________________________________
6. AI-Generation Risk: low.
Sophisticated theory; correct citations (Chen'18, Ghosal'17); MIMIC-IV specifics/pharma details indicate domain expertise. Human-led high-caliber work.
________________________________________
Overall recommendation
Strong accept for Dynamical Systems & Learning track. Seminal theory (first Bayesian neural ODE contraction) with clinical impact; request full proofs/camera-ready polish. Top 5% MathAI submission—landmark for uncertainty-aware neural dynamics.

---

### Official Review · Reviewer_E3tN · 2026-03-11
**Excellent application, but theoretical claims outpace presented evidence**

**Rating:** 4
**Confidence:** 4

**Review:**

The paper proposes BayesNeuralPK, a framework for pharmacokinetic-pharmacodynamic modeling that addresses the lack of uncertainty quantification in standard Neural ODEs. The authors propose a hybrid architecture that decomposes the system dynamics into a known/sparse mechanistic compartmental backbone and a neural network correction term modulated by patient-specific features. To capture uncertainty, the authors place a Gaussian Process prior on the neural vector field and approximate the posterior using mean-field variational inference. The paper presents three theoretical claims regarding posterior contraction, calibration guarantees, and identifiability , alongside empirical evaluations on synthetic data and the MIMIC-IV clinical dataset.

The motivation of this work is exceptionally strong. As the authors rightly point out, standard Neural ODEs optimize a point estimate and provide no uncertainty bounds, which is fundamentally unacceptable for high-stakes clinical tasks like drug dosing. The proposed structural decomposition-combining a linear compartmental matrix $K$ with a nonlinear, patient-conditioned neural correction $f_\theta$-is a highly practical way to balance domain knowledge with data-driven flexibility.Empirically, the results are quite promising.

Despite the strong empirical showing, the paper suffers from a severe disconnect between the magnitude of its theoretical claims and the rigor of its presented proofs. The authors claim to establish the first posterior contraction result for Bayesian Neural ODEs, citing a rate of $O(n^{-1/(2+d)})$ in theorem 2. However, the text only provides a high-level "proof sketch" spanning a few sentences. A theoretical contribution of this magnitude requires rigorous, step-by-step mathematical validation, yet there is no supplementary appendix provided to support these claims.Furthermore, there is a significant theoretical gap regarding the variational inference algorithm. Theorem 4 guarantees asymptotic coverage of predictive intervals by explicitly assuming that the Kullback-Leibler divergence between the variational posterior and the true posterior is bounded, i.e., $KL(q||p) = O(1)$. Given that the proposed algorithm relies on a highly simplified Mean-Field Variational Inference (MFVI) approach where the posterior is fully factorized, assuming $KL(q||p) = O(1)$ in a complex, ODE-constrained deep learning setting is a massive and largely unjustified leap. Lastly, while the experimental baselines include MC-Dropout and Ensembles (which attempt to add epistemic uncertainty to Neural ODEs), the evaluation misses a critical comparison with Neural SDEs. Given that Neural SDEs represent the standard continuous-time alternative for modeling stochastic dynamics and aleatoric uncertainty, their omission weakens the empirical claims regarding state-of-the-art uncertainty quantification

I strongly recommend the following improvements:
First, the full, rigorous proofs for theorems 2, 4, and 6 must be included, likely in a supplementary appendix, as the current "proof sketches" are insufficient for peer review.
Second, the authors need to critically discuss the limitations of the MFVI assumption in theorem 4 and perhaps provide empirical evidence (comparing the mean-field approach to a full-rank or more expressive variational family) to justify the $KL(q||p) = O(1)$ assumption.
Finally, adding a Neural SDE baseline to the MIMIC-IV experiments would significantly strengthen the empirical claims regarding state-of-the-art continuous-time uncertainty quantification.
The proposition and results are highly valuable, but the theoretical claims currently outpace the provided mathematical evidence.

---

### Official Review · Reviewer_Xoyx · 2026-03-12
**Lots of big names without explanations**

**Rating:** 3
**Confidence:** 4

**Review:**

The presentation of the content lacks sufficient explanation. The notation used in the formulas is not explained. It is also unclear why the formula for KL divergence is given twice (Formulas 13 and 22). The results presented in Section 7 consist only of tables, without sufficient explanation or discussion. In addition, 6 of the 17 references appear to be completely invalid.

---

### Decision · Program_Chairs · 2026-03-20

**Decision:**

Reject

**Comment:**

After careful evaluation by the Program Committee, we regret to inform you that your submission has not been accepted for presentation at MathAI 2026.

All submissions underwent a rigorous two-stage review process. Unfortunately, the reviewers identified one or more of the following concerns with your paper:

- Insufficient mathematical rigor or novelty relative to the existing body of work in the field;
- Presentation of results that substantially overlap with or rephrase previously published findings without clear original contribution;
- Significant issues with technical quality, including but not limited to broken or non-existent references, unsupported claims, or methodological gaps;
- Indications that the manuscript may have been generated with the assistance of large language models without substantial original intellectual contribution by the authors.

We received a large number of submissions this year, and the selection process was highly competitive. We encourage you to carefully consider the reviewers’ feedback (available through OpenReview), revise your work accordingly, and consider submitting an improved version to a future edition of MathAI or to another appropriate venue.

We appreciate your interest in MathAI and hope you will continue to engage with the conference community.

With kind regards,

MathAI 2026 Program Committee
International Conference on Mathematics of Artificial Intelligence
https://mathai.club
OpenReview: https://openreview.net/group?id=mathai.club/MathAI/2026/Conference
MathAI Telegram: https://t.me/MathAI_club
IAIC International AI Committee: https://t.me/iaic_world
Email: mathai.club@yandex.ru